# Cyberchondria in the age of COVID-19

**Natasa Jokic-Begic**[ID]*[a], **Anita Lauri Korajlija**[ID][a], **Una Mikac**[ID][a]

Department of Psychology, Faculty of Humanities and Social Sciences, University of Zagreb, Zagreb, Croatia

[a] These authors contributed equally to this work.
* njbegic@ffzg.hr

**Data Availability Statement:** All data files are available from the Croatian Social Science Data Archive (DOI: 10.23669/SIHKQ9).

**Funding:** The author(s) received no specific funding for this work.

## Abstract

The global epidemic of (mis)information, spreading rapidly via social media platforms and other outlets, can be a risk factor for the development of anxiety disorders among vulnerable individuals. Cyberchondria can be a vulnerability factor for developing anxiety in a pandemic situation, particularly when the Internet is flooded with (mis)information. The aim of our study was to examine how cyberchondria is related to changes in levels of COVID-19 concern and safety behaviours among persons living in Croatia during the period in which the first COVID-19 case was identified and when the country recorded its first fatality. Repeated cross-sectional data collection was conducted during two waves over a period of three weeks (N1 = 888; N2 = 966). The first began on the day of the first confirmed case of COVID-19 in Croatia (February 24th, 2020) and the second wave began three weeks later, on the day the first COVID-19 fatality was recorded in Croatia (March 19th, 2020). Participants completed an online questionnaire regarding various COVID-19 concerns and safety behaviours aimed at disease prevention (information seeking, avoidance and hygiene) and a measure of cyberchondria (Short Cyberchondria Scale, SCS). We analysed whether changes to the epidemiological situation during the period between the two waves of data collection led to an increase in COVID-19 related behaviour directly and indirectly via an increase in COVID-19 concerns. The results indicated that, between the two waves of research, there was a pronounced increase in concerns regarding COVID-19 ($b$ = 1.11, $p$ < .001) as well as significant behavioural changes ($b$ = 1.18–2.34, $p$ < .001). Also, results demonstrated that cyberchondria plays a moderating role in these changes. In the first wave, persons with severe cyberchondria were already intensely concerned with safety behaviours. High cyberchondria and high levels of concern about the COVID-19 are associated with intense avoidance behaviours, $R^2$ = .63, $p$ < .001. A moderated partial mediation model was confirmed, in which the effect of the epidemiological situation was weaker for those with higher results on the SCS (as indicated by index of moderated mediation between -.10 and -.15, $p$ < .05). As such, cyberchondria is a contributing factor to long-term anxiety and its impact during pandemic on the general mental health burden should therefore be further investigated.

**Competing interests:** The authors have declared that no competing interests exist.

## Introduction

Throughout its history, humankind has faced infectious diseases. When today's confrontation with infectious outbreaks is put into historical context, we might argue that modern society is in a much better position given the development of a science that searches for causes and methods of treatment, a health system that can provide adequate care to many people and information connectivity that allows for high-speed exchange of information. However, while addressing the physical consequences of infectious diseases is better than ever before, the psychological effects of such outbreaks are widespread and probably more serious. Previous research has demonstrated that a pandemic can have a wide range of psychosocial impacts. On a personal level, people are likely to experience fear for their health, family, safety or finances [1]. On a social level, there is a danger of stigmatization and marginalization of persons who have been in contact with the virus or have been infected [2].

Although infectious diseases have always invoked fear, this response has never been of such a global nature, as is the case for COVID-19 due to global information connectivity. Since the appearance of the first patients in China and Europe, all forms of media have been flooded with information about the spread of the virus and the introduction of measures in various countries. As a result, an imminent threat was perceived in other parts of the world even before a single COVID-19 case was recorded in a given country. Indeed, the World Health Organization declared that we currently face two major threats to our health: the pandemic and an 'infodemic' [3].

The internet has become an essential global source of health information [4], where communication is conducted over giant digital social media platforms capable of sharing information with high transmission speed, reach and penetration. The ability to spread information quickly during the pandemic has proven to have numerous advantages in that it has enabled health systems to prepare for the epidemic and allowed individuals to understand the seriousness of the threat. Incoming information has also served to raise anxiety, which has, in turn, prompted the swift and widespread adoption of safety behaviours propagated by health authorities. From the time that COVID-19 became a global issue in January 2020, universally recommended safety behaviours such as hygiene measures (washing hands, cleaning surfaces), avoidance of social contact, staying at home and wearing a protective mask have been communicated to individuals via mass media. In addition to these individual recommendations, governments worldwide ordered lockdowns to prevent social gatherings and further spread of the virus.

As an unintended consequence, these lock-downs might have unintentionally directed people towards being overloaded with information from social media, information that has very often highlighted COVID-19 as a unique threat [5] and that spreads disturbing images and catastrophic (mis)information about COVID-19 [6]. Experiences with previous and current health crises suggest that repeated media exposure to a community crisis can lead to increased anxiety and heightened stress responses that can lead to downstream effects on health [7, 8]. Fears about diseases, like the diseases themselves, spread through social networks [1]. Search engines and social media platforms further complicate the matter with personalized content, which can contribute to some groups of people receiving better and more accurate information regarding the pandemic situation than others [9]. Furthermore, ubiquitous and constant social media exposure can lead consumers to inaccurately estimate the threat to their own communities [7].

Croatia is a southern European country that shares a sea border with Italy. Since the beginning of February 2020, the COVID-19 crisis has been top news while the country awaited its' first confirmed case. The first case of COVID-19 in Croatia was identified on February 24,

2020. Lockdown was declared on March 16[th], and the first fatality due to COVID-19 was recorded on March 19, 2020. The lock-down measures declared were among the strictest in the EU and included closing schools, shops and all public transport and all social gatherings for more than four people. After lockdown, press conferences of the National Civil Protection Headquarters were held daily, initially twice a day and later once a day. There were constant appeals in the media for citizens to voluntarily implement protective measures such as hand-washing and physical isolation along with the #stay-at-home recommendation. Unfortunately, sensationalist portrayals of the situation, such as the preparation of large sport centres as temporary hospitals, and the broadcasting of disturbing images from hospitals in Italy and Spain were not avoided. These latter forms of information were especially present on social media. To date (May 25, 2020), the number of patients with COVID-19 in Croatia is among the lowest in Europe (542/million inhabitants, fatality rate is 23/million), a trend that has been attributed to the implementation of quarantine-like measures at an early stage. Research examining psychological reactions to the COVID-19 pandemic in Croatia [10] demonstrated a dramatic increase in concern and safety behaviours among participants during the three-week period between the first identified case and the first fatality. Interestingly, there is a discordance between those who are at most risk for serious consequences of the disease and those who are at greatest risk for maladaptive anxiety. Specifically, parents, and mothers in particular, represent the most concerned group, regardless of age. These findings suggest the presence of misplaced health-anxiety and health-protective behaviours and, longitudinal research is certainly justified in order to determine whether more lasting consequences will be present. Previous experiences with public crises that had extensive media coverage, such as an epidemic or terrorist attack, have been demonstrated to leave an unintended psychological burden for people at relatively low risk for direct exposure [7].

Anxiety and insecurity can trigger a compulsive search for information on social media that will further intensify anxiety, creating a vicious cycle of cyberchondria that is difficult to stop [11]. Cyberchondria has been described relatively recently as behaviour characterized by excessive online searching for medical information associated with increasing levels of health anxiety [12, 13]. A pandemic situation, where media is flooded with ambiguous information, is certainly anxiety provoking. Excessive online searching for health information can itself represent a safety-seeking behaviour (e.g., researching whether symptoms are a sign of a viral infection) and, as a result of potentially disturbing information, can trigger or reinforce further safety-seeking behaviour (e.g., further/excessive Internet use) [7, 13–15]. Recent research has found that, during pandemics such as COVID-19, cyberchondria affects people's threat appraisal and motivates people to adopt recommended health measures more promptly [7]. Conversely, it can be a risk factor for overly heightened concern, catastrophizing and social distancing, which all have a pathological influence on mental health [16].

In most cases, the data presented in the scientific literature on the COVID-19 pandemic to date represent data gathered at a single time point during the pandemic which enables understanding the dynamics of change of perceiving threat and adopting safety behaviours. At times of considerable uncertainty, such as the COVID-19 pandemic, we expect people exhibiting high levels of cyberchondria to overestimate the risk of becoming infected with the virus, to be more concerned and to be more prone to voluntarily introducing safety behaviour even before government-enforced measures, which will in turn prolong the period of their worry. We also expect that, while introducing lock-down measures will elevate anxiety and lead to greater behavioural change in most people, those with higher levels of cyberchondria will continue to be more concerned and more prone to safety behaviours.

The present study aimed to explore predictive role of cyberchondria in the changes in concerns and safety behaviours during first few weeks of pandemic. We present data collected in

the general community samples over two occasions: at the very beginning of the presence of SARS-Cov-2 in Croatia (i.e., when the first patient with COVID-19 was diagnosed) and after lock-down was declared. We tested the model with cyberchondria as predictor of safety behaviours, COVID concerns as mediator and wave of collecting data as moderator. We hypothesize the following: (I) higher cyberchondria is directly associated with higher COVID-19 concerns and more pronounced safety behaviour, inlucding gathering information, avoidance and protection; (II) cyberchondria is indirectly associated with safety behaviours via the COVID-19 concerns as mediator; (III) in the second wave of collection of the data concerns, safety behaviours and cyberchondria will be more pronounced; (IV) the wave has a moderation role in the relationship between cyberchondria, COVID-19 concerns safety behaviours, such that in the second wave those relationships will be weaker.

## Materials and methods

This study was approved by Ethical Commitee of Department of Psychology, Faculty of Humanities and Social Sciences, University of Zagreb (Approval number: PSY/20/03/20).

Participants gave their informed consent, and since the research was online by consenting to participate in the study, they entered the next page by clicking the Start bottom.

### Participants

The analysis conducted in this study is based on data from two general population samples. The procedure is described in more detail later in this paper.

**Sample 1.** In the first wave of data collection, a total of $N = 1200$ participants opened the survey. A total of 888 participants answered at least some of the questions regarding relevant characteristics, of which most were female (83.1%) with an age range between 18 and 72 years ($M = 31.3$; $SD = 10.45$). For $N = 888$ participants, there was no missing data, while for all others there was a maximum of one missing item. The partial respondents made 0.9% of the respondent sample, calculated according to Newman [17].

**Sample 2.** In the second wave of data collection, $N = 1320$ participants opened the link to the survey, of which $N = 966$ people from the general population answered at least some of the questions regarding relevant characteristics, and from them most were female (75.8%) with an age range was between 19 and 77 years ($M = 40.0$; $SD = 11.94$). For $N = 925$ participants, there was no missing data, while for others there were one or two missing items, except for two participants who did not answer seven/eight items. The partial respondents made 4.2% of the respondent sample, calculated according to Newman [17].

All participants in both samples were residents of Croatia. Table 1 presents the final sample sizes and the demographic characteristics of each sample.

### Procedure

Repeated cross-sectional data collection was conducted. The first wave (Sample 1) began on the day of the first confirmed case of COVID-19 in Croatia (February 24th, 2020) and the second wave (Sample 2) began three weeks later, on the day the first COVID-19 fatality was recorded in Croatia (March 19th, 2020). Lockdown was declared on March 16th, 2020.

Data were collected anonymously online using a snowball method. Invitations to participate with a link to the online survey on the Survey Monkey web domain was shared on social networks and sent via e-mail to acquaintances. The invitation and link for the first survey was also included in an interview given by one of the authors for a major Croatian online news portal (index.hr). The data in both waves were collected within one week from when they were posted.

**Table 1. Demographic characteristics of Samples 1 and 2.**

| Variable | Sample 1 | | Sample 2 | |
|---|---|---|---|---|
| | *M (SD)* | *N (%)* | *M (SD)* | *N (%)* |
| Age | | | | |
| Men | 33.0 (10.43) | 150 (16.9) | 42.0 (12.09) | 234 (24.2) |
| Women | 30.9 (10.43) | 738 (83.1) | 39.3 (11.83) | 723 (75.8) |
| Total | 31.3 (10.45) | 888 | | 957 |
| Education | | | | |
| Primary school degree | | 3 (0.3) | | 7 (0.7) |
| Secondary school degree | | 240 (27.1) | | 250 (25.9) |
| Bachelors or Graduate degree | | 578 (65.1) | | 628 (65.0) |
| Postgraduate degree | | 67 (7.5) | | 81 (8.4) |
| Children | | | | |
| Yes | | 257 (17.9) | | 492 (50.9) |
| No | | 631 (71.1) | | 474 (49.1) |
| Number of children | | | | |
| One | | 115 (44.7) | | 170 (38.5) |
| Two | | 107 (41.6) | | 219 (49.7) |
| Three to six | | 35 (13.7) | | 52 (5.4) |
| Chronic health condition | | | | |
| Yes | | 159 (17.9) | | 187 (19.4) |
| No | | 729 (82.1) | | 779 (80.6) |

In Sample 1, the completion rate was 75% and in Sample 2, it was 73%. Participants drop out occurred gradually throughout the survey, with no systematic drop-out factor evident. In the second wave, participants were asked if they had participated in the first wave, to which 34.4% answered positively. However, because we could not match the data from both waves, we considered them to be independent samples even though we are aware that some participants overlapped.

## Measures

In both waves, we collected data regarding health concerns related to COVID-19, health behaviours related to preventing the spread of the disease and level of cyberchondria.

At the time the research was planned and conducted, in the scientific literature there were very few papers on the psychological aspects of the COVID-19 pandemic, and they came mostly from China. Instruments measuring fear of COVID-19 become available latter [18], so we used instruments known from previous epidemics in the construction of instruments for this research.

In the light of the changing nature of concerns over time, we used two slightly different sets of items to assess participants' health concerns in relation to the COVID-19 disease.

**The COVID-19 Anxiety Scale**–In the first wave, this measure (CAS-1) was a six-item self-rating scale inspired by the Swine Flu Anxiety Items [19] scale. It assesses participants' concerns about the spread of SARS-Cov-2, perceived likelihood of contracting the virus (themselves and someone they know), perceived severity of infection, concerns about whether the situation will become an epidemic and perceived severity of this virus in relation to flu. Participants rated the extent to which each item related to them on a 5-point scale ranging from 1 (not at all) to 5 (very much).

The COVID-19 Anxiety Scale–In the second wave, this measure (CAS-2) was a nine-item self-rating scale in which five items were the same as in the first wave (the item regarding epidemic was not included in light of the fact that pandemic had been declared by WHO by the time of the second wave). An additional 4 items were also included–two items regarding concerns related to the perceived likelihood of older and younger family members contracting the virus, one item related to concerns about whether one's mental health would worsen in the future and one item related to concerns about worsening mental health of significant others. Participants rated the extent to which each item related to them on a 5-point scale ranging from 1 (not at all) to 5 (very much).

In this paper, the five items included in both waves were analysed (CAS-5 items) [10]. The psychometric properties of the CAS-5 showed promising results [10].

**The COVID-19 Safety Behaviour Checklist** (CSBC) [10] is an eleven-item checklist inspired by the Ebola Safety Behaviour Checklist [20]. It assesses participants' use of safety behaviours designed to prevent contracting the COVID-19 disease (e.g. washing hands, avoiding strangers, avoiding leaving the house). On a 5-point scale ranging from 1 (not at all) to 5 (most of the time), participants in the first wave rated the extent to which they would engage in these activities over the following days due to concerns about COVID-19. In the second wave, participants gave ratings as they related to the past week. So far, the psychometric properties of the CSBS showed promising results [10, 21].

**The Short Cyberchondria Scale** (SCS) [22] measures negative emotional reactions when searching for symptoms online. It consists of four items that measure the negative aspects of cyberchondria (e.g. „*After searching for health information, I feel frightened* "). Participants responded by indicating their agreement to each item using a 5-point scale ranging from 1 (strongly disagree) to 5 (strongly agree). A higher result indicates a higher level of cyberchondria. The Croatian version of the SCS has shown good psychometric properties [22].

## Statistical analysis

First, we show the percentages of indicative answers on specific items to describe the situation in Croatia. As a preliminary analysis, we verified the factor structure of the instruments using parallel and exploratory factor analysis of the data from Sample 1 and confirmatory factor analysis of the data from Sample 2. Model fit of the proposed model was assessed using chi-square, comparative fit index (CFI; > 0.9), root mean square error of approximation (RMSEA; < 0.08) and standardized root mean square residual (SRMR; < .08) [23]. We also verified the measurement invariance of the two samples, in which a change in CFI > .01 and in RMSEA > .015 was considered significant [24]. Structural equation models were estimated with a full information maximum likelihood approach and effects coding identification. The internal consistency of all scales was calculated as McDonald's omega and Cronbach alpha coefficient. These analyses were performed using the R software (packages psych, lavaan, semTools) [25, 26]. Finally, we tested three moderated mediation models, one for each of the behaviour subscales, using PROCESS macro for SPSS [27]. For the indirect effects and their moderation in these models we show the bootstrapped CIs calculated with 5000 samples. Except for confirmatory factor analysis, pairwise method was used to treat missing data, partially due to low percent of partial respondents in the respondent sample [17].

## Results

In light of the fact that this research deals with a new and unique situation in which it is important to examine things at the phenomenological level, our analysis began with an examination of change at the level of each individual concern and behaviour. The data presented in Table 1

presents results related to individual items of the COVID-19 Anxiety Scale, COVID-19 Safety Behaviour Checklist and the Short Cyberchondria Scale.

For the Short Cyberchondria Scale, parallel analysis indicated the existence of one factor composed of all items in Sample 1, which was confirmed by the CFA in Sample 2, $\chi^2(2) =$ 4.352, $p = .113$, CFI = .998, RMSEA = .035, 10%CI [.000, .081], SRMR = .012. The scale showed structure, loading and partial intercept invariance on two occasions, in which two of the items had invariant intercepts (Table 2).

Parallel analysis of CSBC indicated the existence of two factors. However, four of the items had similar loadings on both factors (ranging between .33 and .50). Therefore, in CFA we compared the one-factor ($\chi^2(44) = 741.421$, $p = .000$, CFI = .677, RMSEA = .128, 10%CI [.120, .136], SRMR = .076), the two-factor ($\chi^2(43) = 621.188$, $p = .000$, CFI = .732, RMSEA = .118, 10%CI [.110, .126], SRMR = .070) and the three-factor ($\chi^2(42) = 489.760$, $p = .000$, CFI = .793, RMSEA = .105, 10%CI [.097, .114], SRMR = .058) models, where the latter demonstrated a significantly better fit. These three factors were Avoidance Behaviour, Information Searching and Protection Behaviour. Due to inadequate fit of the three-factor model, we included two correlations between items, $\chi^2(40) = 161.310$, $p = .000$, CFI = .944, RMSEA = .056, 10%CI [.047, .065], SRMR = .039. Further analysis indicated that these items are quite similar in terms of content (e.g., Avoid leaving the house and Avoid places with many people) and had high correlations on both occasions (between .52 and .75). As such, we decided to keep only one item per

**Table 2. Percentage of the participants with high endorsement (answers 4 or 5) of specific items of COVID-19 anxiety (relates much/very much), safety behaviours (often/always) and cyberchondria (often/most of the time) items.**

| Items | 1st wave | 2nd wave |
|---|---|---|
| *COVID-19 anxiety* | | |
| How worried are you about the COVID-19 virus? | 23.1 | 47.9 |
| How likely do you think it is that you will be infected with the COVID-19 virus? | 19.6 | 33.2 |
| How likely do you think it is that someone you know will be infected with the COVID-19 virus? | 37.4 | 64.3 |
| In the event that you become infected with the COVID-19 virus, how worried are you about becoming seriously ill? | 21.6 | 45.4 |
| In your opinion, how much more dangerous is this virus than flu? | 26.3 | 82.1 |
| *COVID-19 behaviours* | | |
| Wash hands more frequently and thoroughly than usual | 61.5 | 88.9 |
| Avoid places with many people | 40.9 | 96 |
| Avoid leaving the house | 10.8 | 77.3 |
| Follow news related to the spread of the COVID-19 virus more frequently | 56.6 | 89.1 |
| Stock up on food and supplies for a crisis situation | 7.9 | 39.2 |
| Wear a protective mask | 3 | 17.6 |
| Use hand disinfectant/sanitizer | 45.4 | 71 |
| Avoid shaking hands with others I | 18.9 | 89.8 |
| Avoid people who look ill | 43.9 | 65.2 |
| Avoid strangers | 19.9 | 54 |
| Search the Internet for information | 48.4 | 76.1 |
| *Short cyberchondria scale* | | |
| After searching for health information, I feel frightened. | 13.2 | 19.9 |
| After searching for health information, I feel frustrated. | 15.2 | 22 |
| After searching for health information, I feel confused by the information that I found. | 14.7 | 13.2 |
| Once I start searching for health information, it is hard for me to stop. | 12.7 | 19.6 |

**Table 3. Comparison of measurement invariance models between Wave 1 (n = 888) and Wave 2 (n = 966).**

| Invariance models | SCS | | | | | CSBC | | | | | CAS | | | | |
|---|---|---|---|---|---|---|---|---|---|---|---|---|---|---|---|
| | $\chi^2$ | df | RMSEA [90% CI] | CFI | ΔCFI ΔRMSEA | $\chi^2$ | df | RMSEA [90% CI] | CFI | ΔCFI ΔRMSEA | $\chi^2$ | df | RMSEA [90% CI] | CFI | ΔCFI ΔRMSEA |
| Equal structure | 19.808 | 4 | 0.065 [0.038, 0.095] | .993 | | 125.509 | 22 | .071 [.058, .084] | .964 | | 27.844 | 4 | .080 [.054, .11] | .981 | |
| Equal loadings | 40.262 | 7 | .072 [.051, .094] | .985 | .008 .007 | 130.280 | 26 | .066 [.055, .077] | .964 | 0 .005 | 44.282 | 7 | .076 [.055, .098] | .971 | .010 .004 |
| Equal intercepts | 95.810 | 10 | .096 [.079, .114] | .960 | .025 .024 | 342.959 | 30 | .106 [.096, .116] | .892 | .072 .040 | 197.431 | 10 | .142 [.125, .160] | .852 | .119 .066 |
| Partially equal intercepts [a] | 43.765[b] | 9 | .065 [.046, .084] | .984 | .001 .007 | 135.786[c] | 28 | .064 [.054, .076] | .963 | .001 .002 | 47.220[d] | 8 | .073 [.053, .093] | .969 | .002 .003 |

*Note*. All models are compared to the model above them in the table, except [a], which is compared to the model with equal loadings. Invariant intercepts: [b] Items 2 & 3, [c] Items 1, 4, 7 & 11, [d] Items 2. SCS = Short Cyberchondria Scale; CSBC = COVID-19 Safety Behaviour Checklist; CAS = COVID-19 Anxiety Scale; SB = Satorra-Bentler; RMSEA = root mean square error of approximation; CI = confidence interval; CFI = comparative fit index. The internal consistency of all scales, expressed by McDonald's omega and Cronbach alpha coefficients, is presented in Table 4. For most scales, the reliability was lower on the second wave (i.e. Sample 2), especially for Protection Behaviour in the COVID-19 Safety Behaviour Checklist.

each correlated item pair in the final version. When testing measurement invariance, multiple items demonstrated loading invariance. Content analysis of the items revealed that items examining behaviour for which there were contradictory recommendations at the time of measurement (Wear protective masks & Stock up on food and supplies for a crisis situation). Therefore, we excluded these two items. The final model for which we tested invariance included three factors: *Avoidance Behaviour*, consisting of three items (Avoid shaking hands with others; Avoid people who look ill; Avoid leaving the house), *Information Searching*, consisting of 2 items (Follow news related to the spread of the COVID-19 virus more frequently; Search the Internet for information) and *Protection Behaviour*, consisting of 2 items (Use hand disinfectant/sanitizer, Wash hands more frequently and thoroughly than usual). All factors demonstrated structure, loading and intercept invariance, except Avoidance Behaviour, which did not show intercept invariance (Table 3).

The COVID-19 Anxiety Scale demonstrated a one-factor structure in Sample 1, as indicated by parallel analysis. However, high collinearity of two items led to an unstable structure in the CFA, $\chi^2(5) = 269.625$, $p = .000$, CFI = .645, RMSEA = .234, 10%CI [.211, .258], SRMR = .092. After the exclusion of one of these two items, the one-factor model showed adequate fit to the data, $\chi^2(2) = 13.046$, $p = .001$, CFI = .973, RMSEA = .076, 10%CI [.040, .117], SRMR = .023.

**Table 4. Pearson correlations and reliability of the Short Cyberchondria Scale (SCS, k = 4), three COVID-19 Safety Behaviour Checklist subscales (CSBC, k = 2/3/2) and the COVID-19 Anxiety Scale (CAS, k = 4) at the two waves (1 & 2).**

| | SCS | CSBC—Information | CSBC—Avoidance | CSBC—Protection | CAS | omega/alpha |
|---|---|---|---|---|---|---|
| SCS | | .338** | .323** | .261** | .371** | .79/.78 |
| CSBC—Information | .214** | | .425** | .460** | .501** | .80/.79 |
| CSBC—Avoidance | .159** | .284** | | .501** | .515** | .74/.72 |
| CSBC—Protection | .207** | .166** | .316** | | .389** | .70/.70 |
| CAS | .373** | .274** | .328** | .258** | | .77/.75 |
| omega/alpha | .79/.77 | .64/.64 | .57/.58 | .41/.40 | .61/.60 | |

*Note*. Data from Wave 1 are above the diagonal (*n* = 881–888) and data from Wave 2 are below the diagonal (*n* = 929–966).

** $p < .001$.

This model demonstrated structure and loading invariance on two waves, but not intercept invariance, where only one item had an invariant intercept (Table 3).

In light of the fact that all scales demonstrated structure and loading invariance, further analysis regarding their relationships is justified. However, the lack of intercept invariance implies that comparing the mean levels of variables across two occasions may produce misleading results.

To establish whether cyberchondria predicted COVID-19 anxiety and behaviour differently over the two measurement waves, we tested a moderated mediation model in which cyberchondria was the predictor, COVID-19 anxiety was the mediator, behaviour was the criteria and wave was the moderator. Three different analyses were performed, one for each of the behaviour subscales (Table 5). Both the direct effects of cyberchondria on behaviour ($b$ = 0.26–0.37) and the indirect effects via anxiety were statistically significant ($b$ = 0.08–0.23), although the indirect effects were somewhat weaker, indicating that cyberchondria is also related to behaviour in a way other than via anxiety. As expected, wave was also a significant predictor, indicating that anxiety and safety behaviours were more pronounced on the second occasion. This was also true for cyberchondria, as indicated by the significant correlation between occasion and SCS, $r$ = .17, $p < .001$.

In general, higher cyberchondria was related to more concerns and more safety behaviour on both waves (Table 5). However, this relationship was stronger in the first wave then on the second occasion, as indicated by the significant interaction between SCS and wave ($b$ = .39 vs. .30 for CAS, .25 vs. .12 for Information Behaviour). Cyberchondria was not related to Avoidance Behaviour in the second wave ($b$ = .19 vs. .03) and was equally related to Protection Behaviour on both waves (insignificant interaction of SCS and occasion: $b$ = -0.06). Generally, most of the effects related to Protective Behaviour are smaller when compared to other behaviours, including this interaction effect, which might be the effect of the smaller reliability of this variable. These moderated relationships are presented in Fig 1. The relationships between anxiety and behaviour have a similar pattern as those for cyberchondria and behaviour (Table 5).

As can be seen in Fig 1, most participants demonstrated high and less variant levels of safety behaviour on Wave 2 (Table 6), which might explain why cyberchondria was less predictive of these behaviours. However, those higher on cyberchondria were already demonstrating higher anxiety and were more prone to safety behaviours on the first wave.

## Discussion

The aim of the present study was to examine how cyberchondria is related to changes in levels of COVID-19 anxiety and safety behaviours among persons living in Croatia during the first phase of the virus outbreak (more precisely, during the period in which the first COVID-19 case was identified and when the country recorded its first fatality). Our results indicate that, in the first three weeks of the outbreak, there was a significant increase in cyberchondria and COVID-19 related anxiety and safety behaviours. In both samples, higher levels of cyberchondria are related to greater concerns and safety behaviours. However, this relationship was stronger at the very beginning of the virus outbreak.

Overall, these results should be interpreted against the context of the situation in Croatia at the time in which the study was conducted. Indeed, it can be argued that this study e captured the unfolding situation during the first few weeks of the virus outbreak, where the first wave of data collection was initiated when the first case of infection was confirmed and, at the outset of the second wave of data collection, there were 105 confirmed cases and the first recorded fatality. In these three weeks, Croatia implemented very high-level restrictions that were among

**Table 5. The effects of cyberchondria (SCS) for predicting COVID-19 related behaviour (CSBC) via COVID-19 related anxiety (CAS), moderated by waves of measurement.**

| Criterium | Predictor | CSBC–Information [a] | CSBC–Avoidance [b] | CSBC–Protection [c] |
|---|---|---|---|---|
| | | *b* | *b* | *b* |
| | | *95% CI* | *95% CI* | *95% CI* |
| CAS | SCS | **0.48** | **0.47** | **0.48** |
| | | **[0.347,0.604]** | **[0.343,0.601]** | **[0.349,0.607]** |
| | Wave | **1.11** | **1.11** | **1.12** |
| | | **[0.924,1.306]** | **[0.914,1.296]** | **[0.926,1.309]** |
| | SCS* Wave (Moderation) | *-0.09* | *-0.08* | *-0.09* |
| | | *[-0.166,-0.007]* | *[-0.164,-0.005]* | *[-0.168,-0.009]* |
| | SCS on Wave 1 (Conditional effect) | **0.39** | **0.39** | **0.39** |
| | | **[0.331,0.448]** | **[0.329,0.446]** | **[0.332,0.448]** |
| | SCS on Wave 2 (Conditional effect) | **0.3** | **0.3** | **0.3** |
| | | **[0.248,0.357]** | **[0.249,0.357]** | **[0.248,0.356]** |
| | | $R^2 = .39$ | $R^2 = .39$ | $R^2 = .39$ |
| | | MSE = .547 | MSE = .547 | MSE = .548 |
| Behaviour | SCS (Direct effect) | **0.37** | **0.34** | *0.26* |
| | | **[0.204,0.541]** | **[0.191,0.484]** | *[0.073,0.447]* |
| | CAS | **0.91** | **0.73** | **0.68** |
| | | **[0.739,1.078]** | **[0.578,0.874]** | **[0.492,0.869]** |
| | Wave | **1.7** | **2.34** | **1.18** |
| | | **[1.342,2.051]** | **[2.033,2.65]** | **[0.784,1.57]** |
| | SCS* Wave (Moderation) | *-0.12* | *-0.15* | -0.06 |
| | | *[-0.228,-0.02]* | *[-0.242,-0.061]* | [-0.177,0.054] |
| | CAS* Wave (Moderation) | **-0.32** | **-0.2** | *-0.21* |
| | | **[-0.437,-0.21]** | **[-0.298,-0.1]** | *[-0.334,-0.082]* |
| | SCS on Wave 1 (Conditional effect) | **0.25** | **0.19** | |
| | | **[0.172,0.325]** | **[0.12,0.252]** | |
| | SCS on Wave 2 (Conditional effect) | **0.12** | 0.03 | |
| | | **[0.054,0.196]** | [-0.027,0.096] | |
| | CAS on Wave 1 (Conditional effect) | **0.58** | **0.53** | **0.47** |
| | | **[0.512,0.658]** | **[0.464,0.59]** | **[0.391,0.553]** |
| | CAS on Wave 2 (Conditional effect) | **0.26** | **0.33** | **0.26** |
| | | **[0.174,0.349]** | **[0.252,0.404]** | **[0.167,0.361]** |
| | Indirect effect of SCS via CAS on Wave 1 [d] | *0.23* | *0.2* | *0.18* |
| | | *[0.181,0.277]* | *[0.16,0.254]* | *[0.139,0.234]* |
| | Indirect effect of SCS via CAS on Wave 2 [d] | *0.08* | *0.1* | *0.08* |
| | | *[0.05,0.111]* | *[0.071,0.131]* | *[0.051,0.112]* |
| | | $R^2 = .36$ | $R^2 = .63$ | $R^2 = .25$ |
| | | MSE = .808 | MSE = .611 | MSE = .994 |
| | Index of moderated mediation [d] | *-.15* | *-.11* | *-.10* |
| | | *[-.207,-.093]* | *[-.161,-.050]* | *[-.168,-.049]* |

*Note.* All bolded effects are significant at $p < .001$. Effects in italic are significant at $p < .05$.

[a] $n = 1819$

[b] $n = 1814$

[c] $n = 1816$

[d] *Bootstrapped* CIs calculated with 5000 samples.

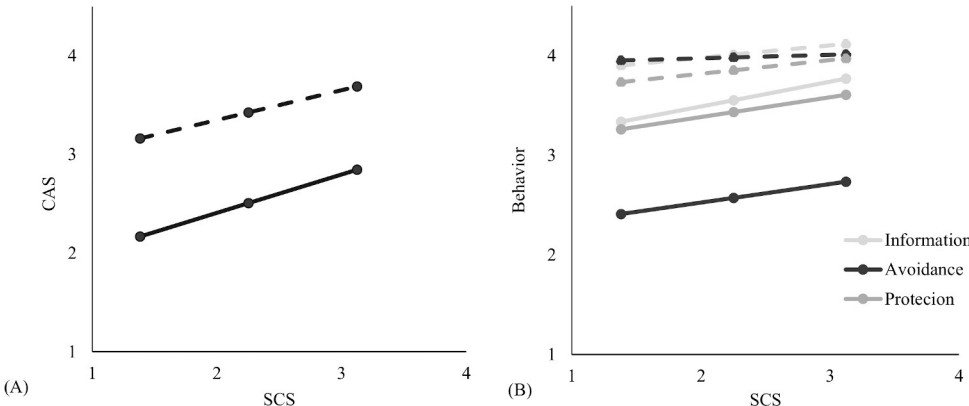

**Fig 1.** The relationship between the Short Cyberchondria Scale (SCS) and a. COVID-19 Anxiety Scale (CAS) and b. the three COVID-19 Safety Behaviour Checklist subscales on Wave 1 (full line) and 2 (dashed line).

the strictest in the EU [28] and achieved the targeted linear epidemic curve. Overall, Croatia had a low number of diagnosed cases: on the 23rd of May (the first day without a single new case), there were 2234 confirmed COVID-19 cases and 99 fatalities. However, the price of this slow spread of the virus was an atmosphere of fear that was fuelled by statements from authorities that were constantly repeated in the media. It seems reasonable to assume that such an atmosphere could contribute to an increase in anxiety and concern among all citizens.

The COVID-19 pandemic is unique in its global nature and in the intensity of the measures introduced. Therefore, it is important to capture and describe the phenomenology of citizens' reaction in different parts of the world in efforts to understand people's experience and behaviour in such extraordinary circumstances. This becomes especially important in the event of a predicted second pandemic wave.

For this reason, our analysis began with a verification of individual items in the first and second waves of data collection for the frequency with which people expressed high levels of concerns and safety behaviours.

For all COVID-19 concerns, the results of this analysis indicate that 'much' or 'very much' responses increased in frequency. In the second wave, more than 82% of participants considered COVID-19 to be a virus more dangerous than flu, as opposed to only 26.3% in the first wave. Participants expressed the lowest concern for becoming infected with the SARS-Cov-2 virus, with around one-third of participants at the second measurement point expressing much or very much concern. However, this does represent an increase in concern from the first wave, where just under 20% said they were concerned. When all items are considered together, 27% of participants in the first wave express moderate to severe anxiety. In the second wave, this percentage jumps to more than 72%, which is consistent with previous studies examining anxiety during the COVID-19 pandemic in other countries [29, 30]. Of course, direct comparison with other research has been hampered using different anxiety measures, ranging from general anxiety measures to pandemic-specific scales [31–33]. In the context of

**Table 6. Means (and standard deviations) of the Short Cyberchondria Scale (SCS), three COVID-19 Safety Behaviour Checklist subscales (CSBC) and the COVID-19 Anxiety Scale (CAS) on the two waves.**

| | SCS | CSBC—Information | CSBC—Avoidance | CSBC—Protection | CAS |
|---|---|---|---|---|---|
| Wave 1 | 2.1 (0.84) | 3.2 (1.18) | 2.3 (1.00) | 3.2 (1.23) | 2.5 (0.88) |
| Wave 2 | 2.4 (0.88) | 4.2 (0.82) | 4.1 (0.75) | 4.0 (0.90) | 3.5 (0.71) |

the COVID-19 pandemic, research by Wang [29] demonstrated that around 30% of participants from the general population had elevated anxiety (as measured by general anxiety measures), while around 75% of participants had elevated levels of COVID-19 related anxiety. In another study conducted at the beginning of the virus outbreak in Germany, authors found that around 50% of participants from the general population expressed moderate to severe COVID-19 anxiety [29].

In the first weeks after the COVID-19 outbreak, the results of the present study also indicate a large increase in safety behaviours. Where, we examined the frequency of various safety behaviours, some of which were issued by authorities after lockdown was introduced. Regardless of whether they were issued officially, all safety behaviours increased significantly. In the second wave, more than 80% of participants reported to avoid handshaking often or constantly, to wash their hands more often than before and to seek information about the COVID-19 disease. Wearing face masks was the least frequently adopted behaviour (3% in the first wave and 17.6% in the second wave), despite having been shown to have an effect in preventing the transmission of infection. Arguably, the wearing of masks is a culturally conditioned behaviour, with previous research indicating it to be far more common in Asian countries than in the West [34].

The results of this study also indicate that, during the first three weeks of the virus outbreak in Croatia, the actual implementation of certain safety behaviours increased. This is especially true of voluntary avoidance behaviours, which rose dramatically from when lockdown was officially introduced. These results are consistent with data from other countries [35] and previous virus outbreaks [36] and indicate that the public was vigilant in its adoption of public health measures.

Information-seeking behaviours also increased significantly, where 46% of participants in the first wave reported to frequently search the Internet for information about COVID-19 and 75% reported doing so in the second wave.

Exposure to social media and excessive online searching during the weeks-long period in which the first confirmed case was expected to appear was associated with an increase in anxiety. As we expected, people exhibiting high levels of cyberchondria at the beginning of the virus outbreak developed higher levels of COVID-19 concern and implemented more safety behaviours. These results are consistent with a number of previous studies examining the role of cyberchondria in the psychological response to the COVID-19 pandemic. A study conducted in Germany with 1615 participants demonstrated that cyberchondria combined with health anxiety is associated with strong virus anxiety [29]. Similarly, data from Finland from a study with 225 participants indicated that cyberchondria influenced one's perceptions of perceived severity and perceived vulnerability for COVID-19and also played a significant role in motivating people to adopt recommended health measures [7]. The results of the present study indicate direct effects of cyberchondria on behaviour and indirect effects via anxiety, although the indirect effects were somewhat weaker, suggesting that cyberchondria is related to behaviour in a way other than via anxiety. It might be argued that cyberchondria prompted the perception of situation severity and thus a readiness to accept recommended safety behaviours. If cyberchondria is conceptualized using a cognitive-behavioural model, compulsive Internet searching might be viewed as a safety-seeking behaviour for reducing health anxiety. Although infectious diseases represent a health-anxiety provoking situation, they are also distinct from other health threats in that the disease can be prevented if one does not come into contact with the source of the infection. In the case of the COVID-19 disease, this is possible by adhering to hygienic measures and avoiding contact with other people. Other health anxiety provoking diseases (such as cancer, cardiovascular disease, neurological diseases, etc.) very often do not have such clear instructions on how to prevent the disease, but rather the only

prevention of a fatal outcome is early observation of symptoms. People who began monitoring the pandemic situation on the Internet early on were likely informed about protection options before others and started applying safety behaviours a few weeks before lockdown, which might explain the direct effect of cyberchondria on behaviour. It would be interesting to examine the role of cyberchondria in adopting harmful behaviours that were propagated during the COVID-19 pandemic (e.g. use of bleach to disinfect surfaces or ingestion of harmful substances such as chlorine dioxide–industrial bleach). It will also be interesting to investigate the role of cyberchondria and COVID-19 anxiety in relaxing safety behaviour, where it might be expected that persons with high levels cyberchondria will continue to engage in safety behaviours longer.

During the second wave, the introduction of lockdown amplified anxiety and safety behaviour in nearly all participants. Again, both anxiety and safety behaviour were higher among those with high levels of cyberchondria. This relationship is higher for COVID-19 anxiety than for safety behaviours, a difference that might be attributed to a decrease in the variance of behaviours in the second wave. Safety behaviour, and avoidance behaviour in particular, has become a social norm. Interestingly, even cyberchondria increased at the second time point. This is an important finding in light of the fact that it is still unclear whether cyberchondria is a behavioural manifestation of health anxiety or whether excessive online searching could encourage the development of excessive health anxiety over time. Previous longitudinal research confirms the hypothesis that excessive searching alone is a risk factor for later onset of mental disorders. A study of healthy participants indicated that, among people low in anxiety, an increase in health-related online searches predicts an increase in anxiety two months later and that an increase in anxiety in this population predicts an increase in online searches. [37]. A second longitudinal study with a general population indicated that increased online searching results in increased depression 12 to 18 month later [38]. Although the present study, in light of its' cross-sectional design, does not allow for causal conclusions, information regarding the growth in cyberchondria is certainly worthy of attention. Arguably, an atmosphere of impending danger might have caused frantic online searching for COVID-19-related information that was perhaps initiated with a wish to gather information and gain a sense of relief (e.g., positive meta-beliefs such as "*I need the Internet research during the pandemic to be better prepared*"). However, such searching more often resulted in the contrary, producing a wealth of (often ambiguous) information and increasing anxiety and distress, which eventually led to cyberchondria [39]. The findings of this study provide unique data on the rate and magnitude of change because it includes two measurement points at the very beginning of the virus outbreak. Data regarding response frequency on individual items of the Short Cyberchondria Scale indicate increased frequencies on items representing the core features of cyberchondria: fear, frustration, and excessive search. This increase may pose a long-term health risk, especially if such behaviour is established and generalized to other diseases and not just for COVID-19.

The implications of our study may contribute to a better understanding of the role of cyberchondria in times of virus outbreaks, where research conducted at the outset of the virus outbreak might be helpful to better understand the development of anxiety. We have placed our research focus on cyberchondria specifically in light of the fact that, during virus outbreaks, a situation arises in which media reports are increasingly consumed and lockdown additionally directs people to consume such sources of information [29]. By collecting data at two time points (albeit with two different samples), this research provides an opportunity to observe change at the societal level. The findings indicate that, in the first weeks of the virus outbreak, there was an increase in anxiety and a change in behaviour. Furthermore, results indicate that cyberchondria is associated with anxiety about the COVID-19 disease and

contributes to the faster adoption of safety behaviours at the very beginning of the outbreak. However, behaviours that are extremely useful for protection against infectious diseases can be potentially detrimental to mental health [29, 39]. This is an important finding in that is offers guidance to health authorities in relation to public communication measures. Namely, in future similar situations, the general population should be informed about the harmfulness of excessive exposure to information and of searching for information from unreliable sources as something that can be as dangerous to one's health as the virus itself. Another important finding to arise from this study is that, even at the very outset of the pandemic, significantly more people turned to the Internet for information than prior to the pandemic. As such, interventions to alleviate anxiety might also be delivered digitally in order to more effectively reach this vulnerable population and prevent the harmful effects of excessive online searching and (mis) information.

A number of limitations should be mentioned when considering the results of this study. Although the study includes two large samples, it is not representative of the general Croatian population with respect to socio-demographic variables. Specifically, the sample included a greater proportion of women and of people with higher education levels than is the case in the general population In addition, the mode of participant recruitment (including via social media) and the online nature of the study was more likely to attract people with a greater affinity for Internet use. The self-selection of participants could have certainly influenced the results. It is also possible that persons with a higher affinity for Internet use and online activities also exhibit higher overall levels of cyberchondria. However, in comparison to participants from a previous study [22], the participants included in this study demonstrated lower levels in the cyberchondria.

The main limitation in this study arises as a result of the cross-sectional design, which does not allow for any conclusions regarding causality to be drawn.

The measures used in this study for anxiety and safety behaviour were constructed during the pandemic for the purposes of measuring these constructs during this period and are not yet validated. Similar challenges are faced by researchers in numerous countries and the use of ad-hoc instruments makes comparison of results obtained in different countries difficult. As such, it is important to strive for greater research collaboration and, as much as possible, for the application of the same instruments across international contexts so that emotional and behavioural responses to the pandemic can be understood in a cross-cultural context. Longitudinal research should also be conducted, which will provide numerous answers about the long-term consequences of this period on mental health and health behaviours.

## Conclusion

This paper presents the results of one of the few studies conducted over two time points that has focused on cyberchondria, anxiety and safety behaviours associated with pandemic. The findings demonstrated that, over three weeks at the outset of the pandemic, there was a significant increase in anxiety and adoption of safety behaviours. Cyberchondria is associated with both anxiety and safety behaviours even prior to the introduction of official measures for preventing the spread of the virus, which poses a potential risk to mental health but also direction for managing people's responses to safety measures. Namely, because people with high levels of cyberchondria readily follow measures for preventing infection, it is possible that they will similarly follow instructions for preventing the development of problems in mental health. As such, it is important that health authorities provide information about the importance of preserving one's mental health during stressful periods such as virus outbreaks and offer clear instructions for the management of stress and anxiety during such times.

## Author Contributions

**Conceptualization:** Natasa Jokic-Begic, Anita Lauri Korajlija.

**Data curation:** Una Mikac.

**Methodology:** Natasa Jokic-Begic, Anita Lauri Korajlija, Una Mikac.

**Software:** Una Mikac.

**Writing – original draft:** Natasa Jokic-Begic.

**Writing – review & editing:** Natasa Jokic-Begic, Anita Lauri Korajlija, Una Mikac.

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
