## [Decision Letter · Decision Letter 0]

23 Oct 2020

PONE-D-20-22292

Cyberchondria in the age of COVID-19

PLOS ONE

Dear Dr. Jokic-Begic,

Thank you for submitting your manuscript to PLOS ONE. After careful consideration, we feel that it has merit but does not fully meet PLOS ONE’s publication criteria as it currently stands. Therefore, we invite you to submit a revised version of the manuscript that addresses the points raised during the review process.

We look forward to receiving your revised manuscript.

Kind regards,

Chung-Ying Lin

Academic Editor

PLOS ONE

Journal Requirements:

2. Please modify the title to ensure that it is meeting PLOS’ guidelines (https://journals.plos.org/plosone/s/submission-guidelines#loc-title). In particular, the title should be "specific, descriptive, concise, and comprehensible to readers outside the field" and in this case it is not informative and specific about your study's scope and methodology.

Reviewers' comments:

Reviewer's Responses to Questions

**Comments to the Author**

1. Is the manuscript technically sound, and do the data support the conclusions?

Reviewer #1: Partly

2. Has the statistical analysis been performed appropriately and rigorously? 

Reviewer #1: No

3. Have the authors made all data underlying the findings in their manuscript fully available?

Reviewer #1: Yes

4. Is the manuscript presented in an intelligible fashion and written in standard English?

Reviewer #1: No

5. Review Comments to the Author

Reviewer #1: This study is interesting. The authors assess the relationships between cyberchondria COVID-19 concern and preventive behaviors among general population living in Croatia. I have some important concerns that the authors should address them.

1- Abstract: Sampling procedure study time should be added to the abstract. More results should be reported in the result section.

2- Introduction: I think that it will be great if the authors can provide some information regarding how Croatia react to the COVID-19 (e.g., government' policies, infected cases, deaths). I believe that this will strengthen the manuscript. Please update your literature review using the following references:

Hashemi, S. G. S., Hosseinnezhad, S., Dini, S., Griffiths, M. D., Lin, C. Y., & Pakpour, A. H. (2020). The mediating effect of the cyberchondria and anxiety sensitivity in the association between problematic internet use, metacognition beliefs, and fear of COVID-19 among Iranian online population. Heliyon, 6(10), e05135.

Ahorsu, D. K., Lin, C. Y., Imani, V., Saffari, M., Griffiths, M. D., & Pakpour, A. H. (2020). The fear of COVID-19 scale: development and initial validation. International journal of mental health and addiction.

Laato, S., Islam, A. N., Islam, M. N., & Whelan, E. (2020). What drives unverified information sharing and cyberchondria during the COVID-19 pandemic?. European Journal of Information Systems, 1-18.

I cannot see any clear stations on specific research questions and/or hypotheses. I would see a model to understand the novelty of the study.

Method: why you changed your anxiety measure in the second wave? How Ebola Safety Behaviour scale could be used for COVID-19? There are lots of related scales. Statistical analysis is missing! What is your justification/ rationale for the sample size provided? Please provide evidence for the validity of all the measures (or instrument) used for both samples. Please report missing rate and how you handled them?

Results: the tables are messy. For example, table 5. You need to provide more information on table 5: confidence intervals , standard errors etc. how did you asses indicted effect (e.g. bootstrapping)? You need to report all statistics in statistical analyses section to understand more about the study.

I would see the above comments first before assessing discussion section. For now, the results are not acceptable.

6. PLOS authors have the option to publish the peer review history of their article (what does this mean?). If published, this will include your full peer review and any attached files.

Reviewer #1: **Yes: **AHP

---

## [Author Response · Author response to Decision Letter 0]

25 Nov 2020

Answer to the reviewer of the manuscript: “Cyberchondria in the age of COVID-19”, PONE-D-20-22292

Dear Editor,

The review we received is very constructive and valuable. We considered each suggestion very thoroughly and revised our paper according to suggestions. Taking into account your remarks, we think we have improved the article. 

Here is the list of the specific reviewers’ suggestions, our explanation and the changes carried out: 

Reviewer 1 comment: 

Abstract: Sampling procedure study time should be added to the abstract. More results should be reported in the result section.

Answer to the reviewer 1: 

We included study time in abstract and specific results. 

Reviewer 1 comment: 

Introduction: I think that it will be great if the authors can provide some information regarding how Croatia react to the COVID-19 (e.g., government; policies, infected cases, deaths). I believe that this will strengthen the manuscript. 

Answer to the reviewer 1: 

All of these information are already included in the Introduction section – from line 97 to line 115 and they include all data reviewer asked. We suppose that reviewer missed it. If additional information are needed, please can you specify which one. 

Reviewer 2 comment: 

Please update your literature review using the following references:

Hashemi, S. G. S., Hosseinnezhad, S., Dini, S., Griffiths, M. D., Lin, C. Y., Pakpour, A. H. (2020). The mediating effect of the cyberchondria and anxiety sensitivity in the association between problematic internet use, metacognition beliefs, and fear of COVID-19 among Iranian online population. Heliyon, 6(10), e05135.

Ahorsu, D. K., Lin, C. Y., Imani, V., Saffari, M., Griffiths, M. D., Pakpour, A. H. (2020). The fear of COVID-19 scale: development and initial validation. International journal of mental health and addiction.

Laato, S., Islam, A. N., Islam, M. N., & Whelan, E. (2020). What drives unverified information sharing and cyberchondria during the COVID-19 pandemic? European Journal of Information Systems, 1-18.

Answer to the reviewer 2: 

As reviewer suggested, we included all three publications in the manuscript. 

Reviewer 1 comment: 

I cannot see any clear stations on specific research questions and/or hypotheses. I would see a model to understand the novelty of the study.

Answer to the reviewer 1: 

As reviewer suggested, at the end of Introduction section, we presented specific research questions and hypothesis. 

Reviewer 1 comment: 

Method: why you changed your anxiety measure in the second wave? How Ebola

Safety Behaviour scale could be used for COVID-19?

Answer to the reviewer 1: 

We changed anxiety measure due to development of the situation. When the first wave was conducted the information about virus and disease were scarce. In the time of the second wave it become clear that older people are more at risk, that mental health problems were also beginning to emerge due to pandemic, and those concerns were the topic of media coverage in this period. So, we wanted to update the anxiety scale in accordance with situation. As we stated in the article, for the purpose of this article we used only the five same items from both scales. 

The Ebola Safety Behaviour scale was found appropriate for this situation because behaviours listed in it were the same as the safety behaviours recommended for controlling the SARS-CoV-2 virus – e.g. washing hands, wearing a mask, avoiding crowded places. 

We included in the manuscript a sentence clarifying this. 

Reviewer 1 comment: 

There are lots of related scales. Statistical analysis is missing! What is your justification/ rationale for the sample size provided? Please provide evidence for the validity of all the measures (or instrument) used for both samples. Please report missing rate and how you handled them?

Answer to the reviewer 1: 

The section Statistical analysis has been added. We commented the missing rate in the procedure section and in the participants’ section in more details and specified the missing data treatment in more details in the statistical analysis section. CAS and CSBS are relatively new scales so we provided references for their reliability, data on the validity are still not available but will be soon. As possible indicators of validity and to show how the scales are related, we have provided in Table 4 the intercorrelations of the measures used. As for the sample size, it was dependent on the methodology used: due to day-to-day changesin the epidemiological situation, we decided in advance to collect responses within one week, and then we closed the collector. 

Reviewer 1 comment: 

Results: the tables are messy. For example, table 5. You need to provide more information on table 5: confidence intervals, standard errors etc. how did you asses indicted effect (e.g. bootstrapping)? You need to report all statistics in statistical analyses section to understand more about the study.

Answer to the reviewer 1: 

We adjusted the format and added some clarifications to Tables 2-5. We also added information on correlations between scales to Table 4, and CI, R2 and index of moderated mediation to Table 5. In order to ease the comparison, we decided to show the three models in one table, and in order not overload the readers with data, we did not include the SE, which can be calculated from the presented CI.

We wish to thank the reviewer for his advices which have helped to improve our paper.

Natasa Jokic-Begic

Anita Lauri Korajlija

Una Mikac

Zagreb, 16th November 2020

---

## [Editor Report · Decision Letter 1]

27 Nov 2020

Cyberchondria in the age of COVID-19

PONE-D-20-22292R1

Dear Dr. Jokic-Begic,

We’re pleased to inform you that your manuscript has been judged scientifically suitable for publication and will be formally accepted for publication once it meets all outstanding technical requirements.

Kind regards,

Chung-Ying Lin

Academic Editor

PLOS ONE

Additional Editor Comments (optional):

I found that the authors have thoroughly and responsively answer all the reviewer's comments. Therefore, I believe that the current format of the manuscript is at the publication standard. Good job!
---

## [Editor Report · Acceptance letter]

9 Dec 2020

PONE-D-20-22292R1 

Cyberchondria in the age of COVID-19 

Dear Dr. Jokic-Begic:

I'm pleased to inform you that your manuscript has been deemed suitable for publication in PLOS ONE. Congratulations! Your manuscript is now with our production department. 

Kind regards, 

on behalf of

Dr. Chung-Ying Lin 

Academic Editor

PLOS ONE